

# Determination of server location in emergency care systems: an index proposal using data envelopment analysis and the hypercube queuing model

Enzo Barberio Mariano[1], Regiane Máximo Siqueira[1], Caio Vitor Beojone[2], Octaviano Rojas Luiz[1,3], João Victor Rojas Luiz[1] and Hugo Henrique dos Santos[1]

[1] Department of Production Engineering, College of Engineering, São Paulo State University (UNESP), Bauru, São Paulo, Brazil
[2] Urban Transport Systems Laboratory (LUTS), École Polytechnique Fédérale de Lausanne (EPFL), Lausanne, Switzerland
[3] Department of Engineering, Institute of Sciences and Engineering, São Paulo State University (Unesp), Itapeva, São Paulo, Brazil

## ABSTRACT

The objective of this article is to propose a new composite index (CI) that helps to determine the most effective location of servers in an Emergency Care System (ECS), using Benefit of the Doubt (BoD)/Data Envelopment Analysis (DEA) and the Hypercube queuing model. The CI proposed was developed in four stages: (1) definition of a number of possible ECS configurations through the application of mathematical partitions and permutations; (2) application of the hypercube queuing model to determine performance parameters for each ECS configuration; (3) application of DEA/BoD to build the CI and generate performance rankings, and (4) evaluation of the rankings obtained to define the best configuration for the ECS analyzed. Data from two real cases from Brazil were used to assess the CI proposal. The results obtained confirm that: (a) the hypercube model could, relatively quickly, determine the configuration parameters generated; (b) the application of an appropriate DEA/BoD model enabled the different configurations to be ranked with good discrimination; (c) a pattern in the relationship between ambulance concentration and configuration effectiveness could be identified; and (d) the CI proposed would benefit ECS managers who are making resource location decisions.

# INTRODUCTION

Among the multiple factors that are necessary for living longer and better, the presence of a quality health system, including an efficient Emergency Care System (ECS), is of fundamental importance. However, ECS are subject to uncertainties, such as the availability and location of services, which may complicate the effectiveness of their design.

Corresponding author
João Victor Rojas Luiz,
joao.rojas@unesp.br

A ECS location problem is to determine the best placement of server units (ambulances, for example) to maximize service coverage and reduce response times, considering the degree of severity and urgency of incidents, as well as the types and number of ambulances to be dispatched (*Brotcorne, Laporte & Semet, 2003*). Proper server location promotes fast response time, efficient service with better resource coordination and higher survival rates, but the challenge of analyzing and ranking different server configurations still persists (*Bélanger, Ruiz & Soriano, 2019*). To offer an overview of existing methods and their limitations, it is suggested to refer to the study conducted by *Li et al. (2011)*, where distinct types of coverage models for ECS planning were examined, encompassing topics such as set covering, maximal covering, maximum expected covering, maximum availability, gradual coverage, and cooperative coverage. Additionally, the paper explores hypercube queuing models as well as dynamic allocation and relocation models.

Although previous research has focused on finding optimal server locations (*Akıncılar & Akıncılar, 2019*; *Erkut et al., 2009*; *Geroliminis, Kepaptsoglou & Karlaftis, 2011*; *Iannoni, Morabito & Saydam, 2008*; *Iannoni & Morabito, 2007*; *Ingolfsson, Budge & Erkut, 2008*; *Kim & Lee, 2016*; *Rajagopalan, Saydam & Xiao, 2008*; *Rodrigues et al., 2018*; *Rodriguez, De la Fuente & Aguayo, 2020*; *Sudtachat et al., 2020*), evaluating each possible alternative scenario according to multiple factors is still challenging. Many of these papers have focused on the location of servers with the aim of either balancing workloads or improving coverage. The lack of a procedure to determine the best configuration for the location of emergency care systems which simultaneously consider internal and external indicators in a composite index has been noted in these works, because there is the difficulty of assigning weights to the indicators to aggregate them in a single performance index. There is also a gap regarding the testing of multiple scenarios to determine how a best-case scenario might be chosen. These are gaps in the existing literature on ECS that this study aims to address.

A possible solution for these gaps lies in the integration of the hypercube queuing model and the Benefit of the Doubt (BoD) approach to Data Envelopment Analysis (DEA) by means of which a composite index (CI) for each analyzed configuration can be constructed.

The hypercube queuing model is a descriptive model in which different emergency system scenarios can be analyzed through the generation of different indicators (*Souza et al., 2015*). DEA/BoD enables multiple measures (*e.g.*, response time and workload) to be aggregated in an index to compare different spatial distribution scenarios for ambulances.

An analysis of the existing models for solving the emergent medical vehicle location problem, compiled in *Bélanger, Ruiz & Soriano (2019)*, reveals that there is a dearth of models that aggregate multiple performance indicators to rank different ambulance location patterns. Furthermore, in the models reviewed by this study, the authors did not identify attempts to determine optimal locations through DEA/BoD. It is noticeable that there is a lack of research on the integration of these models for emergency systems location problems based on more than one objective function (such as coverage, survival, *etc.*), or the trade-off of important performance measures, such as workload and server response time.

Therefore, the objective of the present research is to propose a composite index, built from a combination of the hypercube queuing model and DEA/BoD, with the intention

of ranking different emergency care system location problem configurations. The proposal of this work, therefore, is to exploit the capacity of the hypercube model to realistically describe spatially distributed systems and the capacity of the Benefit of the Doubt (BoD) approach to DEA to synthesize multiple indicators in a single composite indicator.

Data from previously published actual cases were used to illustrate the use of the method presented in this article (*Beojone, Máximo de Souza & Iannoni, 2020*; *Souza et al., 2015*). These illustrative cases, from Ribeirão Preto and Bauru, two medium-sized cities located in São Paulo state, relate to the publicly funded Brazilian mobile emergency medical service (in Portuguese, "Serviço de Atendimento Móvel de Urgência" or SAMU). These illustrative cases were adopted to validate and refine the proposal to integrate the hypercube queuing model and DEA/BoD.

The results contribute to a better understanding of the spatial distribution of service vehicles in emergency systems. By classifying different configurations of servers, significant patterns emerged. The research also advances theory on the integration between queue models and DEA/BoD by building a specific solution to spatial resource location problems.

The following section, Literature Review, presents theoretical aspects relevant to the techniques hypercube model and DEA/BoD as applied in this research. Subsequently, the proposed composite index is introduced, encompassing the determination of analysis units, inputs, and outputs derived from the hypercube model, as well as the index construction using the DEA/BoD model. In the Results section, the evaluation of the proposed model through the analysis of illustrative cases is presented. Finally, the research conclusions are provided.

## LITERATURE REVIEW

### ECS location problem and Hypercube queuing model

A common goal in managing ECS location problems is optimizing the coverage of a given location, such as a city. An area is considered covered when it is located less than a critical distance from at least one of the existing servers. Location problems based on cluster coverage are prescriptive and their objective is to find the optimal location for server units, in order to guarantee total coverage and the maximization of performance measures (*Brotcorne, Laporte & Semet, 2003*).

An important advance in the theory related to ECS location problems was the proposal of probabilistic coverage models, which consider randomness in server availability as a significant factor. The advantage of these models is that they consider emergency call uncertainties, such as when a call enters the system and its location and duration (*Swersey, 1994*).

The first probabilistic ECS location problem solutions focused on maximizing system coverage, considering the probability of a given server unit being busy. *Daskin (1983)* presented a pioneering study in which location models were extended considering this possibility. *ReVelle & Hogan (1989)* presented a model whose objective was to maximize population coverage within a response time established by a level of reliability. Recent research has sought to relax the three simplifying assumptions presented in previous studies,

namely: server independence, identical server occupancy rates, and server occupancy rate invariant to location.

As a way of relaxing these assumptions, some studies have used the hypercube queuing model to calculate the probabilities of server occupation. The hypercube utilizes spatially distributed queues in probabilistic location scenarios. The queuing system performance indicators offered by the hypercube model can be used in prescriptive models to assess the optimal location of a server (*Brotcorne, Laporte & Semet, 2003*).

From the metrics produced by the hypercube model, it is feasible to aggregate inputs and outputs into a composite index. This process can be accomplished through the implementation of Data Envelopment Analysis (DEA).

## Data envelopment analysis (DEA) and benefit of the doubt (BoD) approach

DEA is a technique developed by *Charnes, Cooper & Rhodes (1978)* to determine the efficiency of each unit under evaluation (the so-called decision-making unit - DMU) by aggregating the multiple inputs (raw materials, energy, capital, *etc.*) and outputs (products, income, customers served, *etc.*) of a system. Since its inception, the field of application for DEA has expanded, with new models and applications of the technique appearing every year (*Liu et al., 2013*).

For example, recent efforts have been made to evaluate the efficiency of transport systems using DEA in combination with other techniques. *Taletović & Sremac (2023)* utilized Principal Components Analysis (PCA), while *Blagojević et al. (2020)* employed fuzzy Analytic Hierarchy Process (AHP). However, these methodologies are primarily employed to streamline the quantity of input and output factors or to determine the optimal prioritization of metrics.

One of the most relevant DEA applications is in the construction of composite indexes (CIs) to measure phenomena considering multiple dimensions (*Cherchye et al., 2006*; *Mariano, Ferraz & Gobbo, 2021*). A composite index (CI) is an indicator that aggregates several independent variables into a single measure. According to *Booysen (2002)*, the construction of a CI involves five steps, which are: (1) indicator selection (redundancy must be avoided and data availability ensured); (2) standardization (a set of procedures to group indicators under the same unit of measurement); (3) weighting (how to determine the degree of importance of each indicator); (4) aggregation (methods for combining indicators into a single measure); and (5) validation, both internal (index consistency) and external (comparison with other indicators). DEA provides solutions for the normalization, weighting, and aggregation steps of the indicators.

The application of DEA for the construction of composite indices is called benefit of the Doubt (BoD), as systematized by *Cherchye et al. (2006)*. There are several different models of BoD and *Mariano, Ferraz & Gobbo (2021)* have systematized and compared most of these models, using them to recalculate the Human Development Index (HDI). *Mariano, Sobreiro & Rebelatto (2015)* states that DEA/BoD can be used to build CIs in two ways: (a) in CIs composed only of desirable attributes (in this case, a constant input equal to 1 is

adopted for all DMUs); and (b) in CIs expressed as a ratio of desirable and undesirable indicators, being the approach used in this work.

The measures within this approach are treated as outputs when they contribute positively to ECS performance and as inputs when they are undesirable for ECS performance.

To measure the performance of an ECS, two metrics are considered:
1. Response time, considering that a quick response can save many lives; and
2. Workload, which may compromise service quality due to employee stress or burnout or equipment overload *etc.*

Response time is considered a desirable indicator because it is a core characteristic of an ECS; workload, on other hand, is considered an undesirable indicator because it is an unavoidable negative consequence of an ECS.

## PROPOSED COMPOSITE INDEX

The index proposed in this study integrates the hypercube model and DEA/BoD to evaluate the location of servers in emergency care systems. Four vital steps are required to build the index: (a) enumerating all possible configurations for the ECS studied (DMU determination); (b) applying the hypercube queuing model to determine performance indicators (inputs and outputs) for each configuration; (c) using DEA/BoD to build a composite index in order to rank different ECS configurations (CI construction); and (d) evaluating the CI of each configuration and the ranking thus obtained to determine the best solution for the real problem (validation). Figure 1 summarizes the four stages of CI construction.

At the conclusion of the four steps, the configurations can be ranked according to their performance. From this classification, an analysis of whether there are clear patterns between the best and worst configurations presented can be made, which will guide the future actions of the emergency system decision makers. It is important to note that the implementation of any solution suggested by the model must be analyzed in terms of its technical and economic feasibility. For example, in the case of a reallocation of servers, the potential costs of adapting infrastructure and expanding staff should be considered.

### Enumeration of configurations (DMU determination)

All location patterns must be compared to define the best configurations. These configurations are defined from a list of all the distribution possibilities of the 'n' servers in the 'A' atoms, which are spatial subdivisions of the region studied. Using the concept of mathematical partition, all permutations for the system studied can be defined.

Firstly, all partitions of the number of servers 'n' must be obtained. For example, if there are $n = 4$ servers available, five partitions can be obtained: "4", "3 + 1", "2 + 2", "2 + 1 + 1" and "1 + 1 + 1 + 1". These partitions have a number of terms equal to 1, 2, 3 and 4, respectively. It should be noted that the number of servers used in this calculation includes only those that can be allocated to different atoms. Thus, considering the previous example, if the number of atoms were $A = 3$, the partition "1 + 1 + 1 + 1" could not be applied to the problem in question.

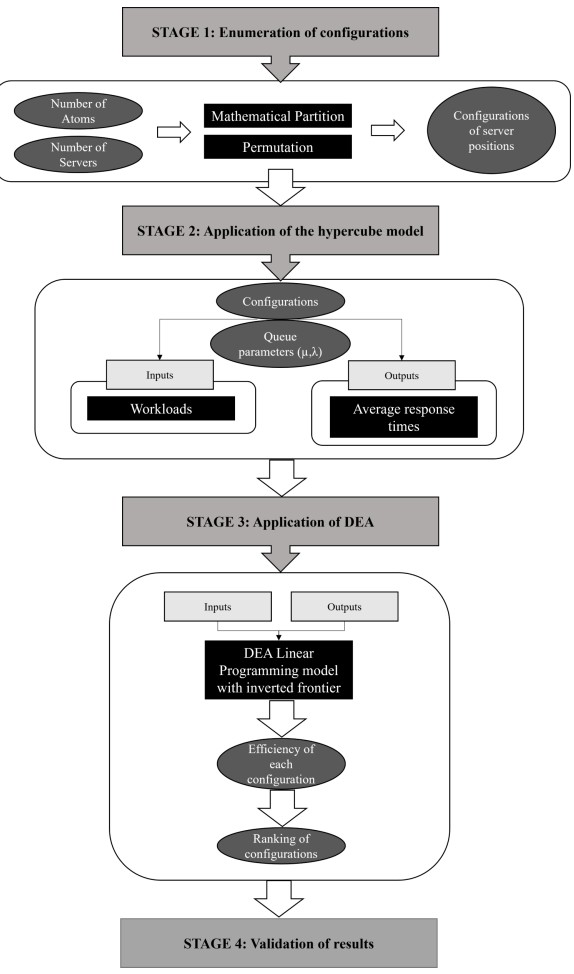

**Figure 1 Integration of the hypercube model with DEA, for ECS location problems.**

The remaining 'P' feasible partitions must be filled with zeros, so all partitions have a number 'A' of terms. So that, in the provided example, the following four adjusted partitions can be observed: P1 = "4 + 0 + 0", P2 = "3 + 1 + 0", P3 = "2 + 2 + 0" and P4 = "2 + 1 + 1".

The total number of configurations can be obtained by permuting the terms of the filled partitions. For instance, in the case of partition P1, three potential configurations are present: "4 + 0 + 0", "0 + 4 + 0" and "0 + 0 + 4". Each permutation, that is, each distribution of ambulances to an atom, will be a different DMU, to be evaluated by DEA in later stages.

The number of configurations (permutations) for each adjusted partition '$P_k$' can be obtained according to Eq. (1). The total number of configurations of a problem is the sum of the configurations generated by all partitions $P_k$, as shown in Eq. (2).

$$C_k = \frac{A!}{\prod_{\forall j} R_j!}. \tag{1}$$

**Table 1  Example of the enumeration of configurations involving two types of servers.**

| Server type | Partition (permutations) | Configuration number (DMU) | Atoms | | |
|---|---|---|---|---|---|
| | | | A1 | A2 | A3 |
| BSU | "4+0+0" (3) | 1 | 4 | 0 | 0 |
| | | 2 | 0 | 4 | 0 |
| | | 3 | 0 | 0 | 4 |
| | "3+1+0" (6) | 4 | 3 | 1 | 0 |
| | | 5 | 3 | 0 | 1 |
| | | 6 | 0 | 3 | 1 |
| | | 7 | 1 | 3 | 0 |
| | | 8 | 1 | 0 | 3 |
| | | 9 | 0 | 1 | 3 |
| | "2+2+0" (3) | 10 | 0 | 2 | 2 |
| | | 11 | 2 | 0 | 2 |
| | | 12 | 2 | 2 | 0 |
| | "2+1+1" (3) | 13 | 2 | 1 | 1 |
| | | 14 | 1 | 2 | 1 |
| | | 15 | 1 | 1 | 2 |
| ASU | "1+0+0" (3) | 1 | 1 | 0 | 0 |
| | | 2 | 0 | 1 | 0 |
| | | 3 | 0 | 0 | 1 |

$$C = \sum_{k=1}^{P} C_k \qquad (2)$$

where:

$C_k$: number of configurations of the partition "$P_k$";

$C$: total of configurations;

$P$: quantity of feasible partitions;

$A$: number of atoms;

$R_j$: number of repetitions of the term j in the partition.

For the case with $n = 4$ basic servers and $A = 3$ atoms, the adjusted partition $P_4 = $ "2 + 1 + 1", generates a number of permutations equal to $\frac{3!}{2!} = 3$; in this case, $R_1 = 2$ because the term "1" is repeated twice. The adjusted partition $P_2 = $ "3 + 1 + 0", on the other hand, generates a number of permutations equal to $3! = 6$ because it does not include repetitions in the terms of the partition. All the 15 configurations generated in this hypothetical case are expressed in Table 1.

If the system consists of distinct types of servers, *e.g.*, Advanced Server Units (ASU) and Basic Server Units (BSU), which differ with respect to the equipment and staff sent to the call, the enumeration must be performed separately for each type of server and, at the end, the results must be multiplied. If, for example, in addition to the four BSUs there was one ASU, three configurations would be generated (see Table 1). The total number

**Table 2  The number of configurations across different numbers of servers and atoms.**

| Atoms | Servers | | | | | | | |
|---|---|---|---|---|---|---|---|---|
| | 3 | 4 | 5 | 6 | 7 | 8 | 9 | 10 |
| 2 | 4 | 5 | 6 | 7 | 8 | 9 | 10 | 12 |
| 3 | 10 | 15 | 21 | 27 | 36 | 45 | 55 | 69 |
| 4 | 20 | 35 | 56 | 84 | 120 | 165 | 220 | 292 |
| 5 | 35 | 70 | 126 | 210 | 330 | 495 | 715 | 1,011 |
| 6 | 56 | 126 | 252 | 492 | 792 | 1,287 | 2,002 | 3,018 |
| 7 | 84 | 210 | 462 | 924 | 1,722 | 3,003 | 5,005 | 8,029 |
| 8 | 120 | 330 | 792 | 1,716 | 3,480 | 6,435 | 11,440 | 19,476 |
| 9 | 165 | 495 | 1,287 | 3,003 | 6,651 | 12,870 | 24,310 | 43,794 |
| 10 | 220 | 715 | 2,002 | 5,005 | 12,160 | 24,310 | 48,620 | 92,423 |

of configurations to be evaluated, therefore, would be 45, equivalent to 15 multiplied by three.

As the number of configurations grows exponentially with the number of servers and atoms, the problem may become computationally unviable. Table 2 shows the configurations resulting from different numbers of atoms and servers. For example, in the situation with 10 atoms and 10 servers the number of configurations is 92,423. This computational constraint might not make applications in real problems unfeasible, since, even for large cities, the problem could be divided into service regions, with fewer atoms. This subdivision of large cities is to be expected, as an ambulance would hardly answer a call in a place very far from its atom.

Based on the definition of possible permutations, performance measures are then calculated. Each permutation represents one possible configuration of the distribution of servers in an atom. The hypercube model will be applied for each configuration, determining its performance measures. A description of how the hypercube model should be built can be seen in the subsequent step.

## Inputs and outputs determination

The second stage involves the application of the hypercube model for each of the configurations built in the previous stage to determine the response times (outputs) and the workloads (inputs).

The original hypercube model is a descriptive model used as a tool for the analysis and planning of urban emergency systems (*Larson, 1974*). In addition to considering uncertainties regarding the origin of calls, service times and the availability of servers, the model addresses the geographic and temporal complexities of a region, based on spatially distributed queues. Basically, the idea is to expand the state space of an M/M/m queue system to represent each server individually, which may include more complicated dispatch policies.

The solution of the model is initiated by the construction of a set of equilibrium equations for the system. These equations describe, in terms of probabilities, the transition between states that have reached a steady state (*Larson, 1974*; *Larson & Odoni, 2007*); that is, the

probabilities are independent of time (*Shortle et al., 2018*). For the hypercube model, a state describes the status of given servers. For example, the state {110} of a system with three servers represents the situation where two servers are busy, and one is free. An equilibrium equation is established for each state transition (for example, {001} to {011}). It is important to mention that the number of states of the hypercube model grow exponentially according to the number of servers, in the order of 2n.

The results of the equilibrium equations are based on the values of the steady-state probabilities of the system, allowing for the calculation of performance measures, such as: server workload, average response time of the system or of each server, and dispatch frequencies, among others (*Larson & Odoni, 2007*).

To apply the classic hypercube model, nine critical assumptions must be verified, as described by *Larson & Odoni (2007)*. Extensions for relaxing these hypotheses, providing a more accurate representation of reality, are found in the literature. Most of these extensions are related to the dispatch policy and operational characteristics of care systems, as can be seen in the work of *Takeda, Widmer & Morabito (2007)*, *Souza et al. (2015)* and *Chiyoshi, Iannoni & Morabito (2011)*.

In our proposal, focused on problems of the localization of emergency systems, the extension presented by *Beojone, Máximo de Souza & Iannoni (2020)* was used (see Appendix). The proposed extension considers the aggregation of servers to reduce the number of equilibrium equations necessary to represent the queuing system; it is especially important in large systems with multiple servers of the same type in one place and allows increasing computational efficiency. The use of servers fully dedicated to answering calls with high priority (where there is a risk of death for the patient) is also another important aspect to be considered in this model.

To use the server aggregation approach, Eq. (3) must be applied to calculate the number of states $|M|$ for a system with T groups of servers. These groups are the partition terms that were defined in the previous section. Servers belonging to the same group must be homogeneous and indistinguishable; that is, they serve the same type of occurrence and are positioned within the same atom.

$$|M| = \prod_{t=1}^{T}(n_t + 1) + |Q| \tag{3}$$

Each group $t$ has $n_t$ servers; $|Q|$ represents the number of queue states; that is, all possible combinations of server occupancy and queue size. It is noted that the number of equilibrium equations in the server aggregation model is significantly less, especially in large systems, than the number of equilibrium equations in the exact hypercube model, as presented by *Beojone, Máximo de Souza & Iannoni (2020)*.

As in this model, if it is considered that there are servers dedicated to calls with high priority in emergency systems, queues can form, even with servers available (*Iannoni, Chiyoshi & Morabito, 2015*; *Rodrigues et al., 2017*). This change in the configuration of the exact hypercube model requires important adjustments to the equilibrium equations and

to the queue. Thus, Eq. (4) calculates the number of queue states:

$$|Q| = |Q_{ds}| + |Q_{pr}| = n_{ds}Q_l + \binom{2 + Q_l}{Q_l} - 1 \tag{4}$$

where:

- $|Q_{ds}|$ is the number of queue states that arise from the model due to fully dedicated servers;
- $|Q_{pr}|$ is the number of queue states in saturated circumstances; that is, when all servers are busy and there are priority levels in the queue;
- $n_{ds}$ is the size of the group of fully dedicated servers;
- $Q_l$ is the queue capacity.

One way to differentiate users and represent priorities in the hypercube model is to subdivide atoms into sub atoms ("layering"). This approach was used by *Takeda (2000)* and *Iannoni (2005)*. Following the strategy presented in *Larson & Odoni (2007)*, sub atoms are created by combining each geographic atom with different types of calls in terms of priority. The integrated model presented here adopts two levels of priority for regular and serious calls. Thus, the number of sub atoms will be double the number of atoms. The Appendix presents the equations referring to the generalization of the state-to-state transitions of the hypercube model (*Beojone, Máximo de Souza & Iannoni, 2020*) proposed in this integration.

Various internal and external performance measures can be calculated from the solution obtained, usually by some numerical method for the solution of linear equation systems (*e.g.*, Gauss-Siedel). These measures include workload and response times, for each atom, server and region, and dispatch frequencies. The workload is calculated according to Eq. (5) (*Souza et al., 2015*).

$$\rho_{it} = \sum_{r \in M} \frac{n_{tr} \cdot P_r}{n_t} \tag{5}$$

where:

$\rho$ is the workload calculated for server $i$ in group $t$;

$r$ is a particular state in the set of states $M$;

$n_{tr}$ is the number of servers in group $k$ occupied in state $r$;

$P_r$ is the probability of the state r.

Workload and average response time are the performance measures used in the DEA model proposed. The average response times for each atom and server were obtained by calibrating the original response time for each configuration. The calibrating procedure can be viewed in *Chiyoshi, Galvão & Morabito (2001)* and *Galvão & Morabito (2008)*.

## Composite index determination

In order not to make the review too extensive, this section will focus on describing the specific elements of the model adopted in this research. For basic definitions of DEA, it is recommended an introductory bibliography on the subject (*Cooper, Seiford & Tone, 2007*; *Cooper, Seiford & Zhu, 2011*).

The inputs for the model will be the workloads of each server. It is important for servers to be able to perform their functions effectively, without excessive workload. The inputs will be represented as $\rho_{ik}$, where the index $k$ will represent each configuration (DMU) and $i$ each server in this configuration; The number of inputs "I" is equal to the number of servers "n".

The outputs of the model, on the other hand, are the reciprocal response times for each server to each sub atom, as one of the main objectives of an emergency system is to promote rapid response (*Halkos & Petrou, 2019*). The use of the reciprocal, as suggested by *Golany & Roll (1989)*, was possible because the variables used did not present null values and because there was no need to maintain the proportionality of the scales for the purpose of this study (*You & Yan, 2011*). As it is an ECS location problem that deals with lives, the adoption of the reciprocal of the average response time of the servers can be justified, because it penalizes more longer response times.

In the present approach, for each configuration "k", three groups of outputs related to the inverse of the average response times were adopted: (a) the inverse of the average response time for an atom $h$ related to regular calls, which is represented by $y_{hk}$; (b) the inverse of the average response time for an atom $h$ related to serious calls, which is represented by $y_{hk}^S$; and (c) the inverse of the average response time of a server $I$, which is represented by $z_{ik}$. The model considers $A$ atoms and $n$ servers. Therefore, the number of outputs "O" in the model is equal to two times the number of atoms (as they are divided into two sub atoms - regular and serious calls) plus the number of servers: O = 2A + n.

There are several models related to Data Envelopment Analysis (DEA), which, according to *Mariano, Sobreiro & Rebelatto (2015)*, vary according to the establishment of different assumptions, such as the type of scale return, the type of efficiency measure (radial or non-radial), and the type of orientation (for input, output or both), among others. As the configuration for this integration proposal is to determine the efficiency of emergency services systems, the orientation to the output, that is, to increase the reciprocal of the response time to the user, is preferable. This is due to the social character of this type of system, in which the priority is to provide a better service level even at the expense of a higher input utilization rate. The primary objective of a health system is to guarantee quality of life through more treatment and assistance. Therefore, from a moral point of view, for this type of system the orientation to the output seems more adequate (*Stefko, Gavurova & Kocisova, 2018*).

In addition, the BCC model (*Banker, Charnes & Cooper, 1984*) seems to be the most appropriate choice, since it is expected that it will become increasingly costly, in terms of additional workload, to decrease the response time by keeping the other characteristics of the system constant. This premise is supported by the healthcare queuing literature (*Palvannan & Teow, 2012*).

To improve the model, the addition of weight constraints, which establish limits on the weighting of inputs and outputs, was necessary (*Cherchye et al., 2006*). The addition of weight constraints introduces subjectivity to the model and should be justified. In the case of emergency systems, it is usual to prioritize calls with respect to their severity and urgency. It makes sense, then, to include in the model the constraint that, for each atom,

the DEA must give greater weight to the response times for the most urgent calls. This type of constraint is called the assurance region and was initially proposed by *Thompson et al. (1986)*. This constraint is expressed in Eq. (12).

To prevent the model from assigning excessive weights to the outputs of a certain group and ignoring another, additional constraints can be inserted by establishing a minimum contribution limit for the efficiency of a given set of weights. This additional constraint considers that the various outputs can be divided into two distinct groups: (a) those referring to the inverse of the response time to atoms (both normal and serious calls) and (b) that of the inverse of the average response time of servers. If the DEA model assigned 100% weight to one group or another, efficient scenarios, in which response times for servers or atoms would be critically high, could be suggested as efficient configurations (*Pedraja-Chaparro, Salinas-Jimenez & Smith, 1997*).

Thus, in order to avoid the model attributing an excessive weight to the outputs of a certain group and ignoring another, it was established that the set of variables linked to the inverse of the response times to atoms should contribute with at least P% of the efficiency, while the variables related to the inverse of the response times of each server should also contribute with at least P% of efficiency, leaving the model free to allocate the missing (100-2P)%. This approach of attributing constraint to the relative contribution of groups of variables was used by *Morais & Camanho (2011)*, inspired by the work of *Wong & Beasley (1990)*.

An advantage of including constraints is to reduce the number of efficient tied DMUs, improving model discrimination (*Wang, Luo & Liang, 2009*). These constraints are expressed in Eqs. (9), (10) and (11). If a user chooses to reduce the subjectivity of the model, it is possible to omit restrictions Eqs. (9), (10) and (11), considering that this may result in the omission of some variables and an increase in the number of ties. To make the determination of P more objective, multicriteria decision-making methods can be applied (*Kuo, Wang & Tien, 2010*).

For each configuration "k" generated in the first step (C configurations in total), the linear programming problem related to the output-oriented BCC model with weight constraints must be solved. Equations (6)–(17) present a possible model in the multiplier form, taking two levels of urgency (normal and serious) and limiting the contribution to the efficiency of each group of outputs by at least P%.

$$\text{Min} \sum_{i=1}^{n} \gamma_i \rho_{i0} - w \tag{6}$$

Subject to

$$\sum_{h=1}^{A} \alpha_h y_{h0} + \sum_{h=1}^{A} \alpha_h^S y_{h0}^S + \sum_{i=1}^{n} \beta_i z_{i0} = 1 \tag{7}$$

$$\sum_{h=1}^{A} \alpha_h y_{hk} + \sum_{h=1}^{A} \alpha_h^S y_{hk}^S + \sum_{i=1}^{n} \beta_i z_{ik} - \sum_{i=1}^{n} \gamma_j \rho_{ik} + w \leq 0, k = 1, 2, \ldots, C \tag{8}$$

$$\sum_{h=1}^{m} \alpha_h y_{h0} \geq 0,01P \tag{9}$$

$$\sum_{h=1}^{A} \alpha_h^S y_{h0}^S \geq 0,01P \tag{10}$$

$$\sum_{i=1}^{n} \beta_i z_{i0} \geq 0,01P \tag{11}$$

$$\alpha_h^S - \alpha_h > 0, h = 1,2,\ldots,A \tag{12}$$

$$\alpha_h \geq \varepsilon \tag{13}$$

$$\alpha_h^S \geq \varepsilon \tag{14}$$

$$\beta_i \geq \varepsilon \tag{15}$$

$$\gamma_i \geq \varepsilon \tag{16}$$

w without sign restrictions $\tag{17}$

where:

- $\gamma_i$ is the weight for the workload of server i;
- $\alpha_h$ is the weight for the inverse of the response time for regular calls in an atom h;
- $\alpha_h^S$ is the weight for the inverse of the response time for serious calls in an atom h;
- $\beta_i$ is the weight for the inverse of the response time for a server i;
- $\rho_{i0}$ isthe workload of server i for the DMU under analysis;
- $\rho_{ik}$ isthe workload of server i for a DMU k;
- $y_{h0}$ is the inverse of the response time for regular calls in an h atom, considering the DMU under analysis;
- $y_{h0}^S$ is the inverse of the response time for serious calls in an h atom, considering the DMU under analysis;
- $y_{hk}$ is the inverse of the response time for regular calls in an atom h, considering a DMU k;

- $y_{hk}^S$ is the inverse of the response time for serious calls in an atom h, considering a DMU k;
- $z_{i0}$ is the inverse of the response time for a server i for the DMU under analysis;
- $z_{ik}$ is the inverse of the response time for a server i, considering a DMU k;
- w is the scale factor;
- P is the minimum contribution to efficiency from each group of response times;
- $\varepsilon$ is a non-Archimedean constant.

The values of the objective function are the efficiencies of each configuration. Since many configurations can result in the same efficiency values, it is necessary to conduct a last discrimination step to obtain a final efficiency ranking.

To deal with cases with many ties between configurations, there are several objective methods to differentiate them, such as the inverted frontier, cross-evaluation, or the triple index (*Mariano, Sobreiro & Rebelatto, 2015*), depending exclusively on the configuration studied. Of all these methods, the inverted frontier stands out for its simplicity, since it consists of just conducting the analysis using outputs as inputs and vice versa. The first to identify the properties of the inverted frontier was *Yamada, Matui & Sugiyama (1994)*. Years later, *Leta et al. (2005)*, *Zhou, Ang & Poh (2007)* and *Mariano & Rebelatto (2014)* proposed three different forms to use the inverted frontier as a discrimination method (*Mariano, Ferraz & Gobbo, 2021*).

The *Leta et al. (2005)* approach is the simplest of these methods; it is based on the construction of a composite index that is the normalized mean between the result of the standard frontier and one minus the result of the inverted frontier. It is important to mention that the inverted frontier proposed by *Leta et al. (2005)* was adopted in this research, but other approaches can also be applied in this step.

## RESULTS

To assess the model, data from two emergency systems were used: SAMU-Bauru and SAMU-Ribeirão Preto. These cases have been described in *Beojone & Souza (2017)* and *Souza et al. (2015)* and were revisited in *Beojone, Máximo de Souza & Iannoni (2020)*. These illustrative cases were used only to validate and refine the proposal to integrate the hypercube model with DEA.

SAMU-Bauru is an emergency medical service (EMS) that serves an urban area of 69 km$^2$ with about 360 thousand inhabitants, located 330 km from the city of São Paulo. Every month, the system responds to about 1,800 calls, classified into four different priorities. The system is divided into six regions (with each region representing an atom for the hypercube model), with only the Boulevard region not having an ambulance allocated to it, as shown in Fig. 2A.

SAMU-Ribeirão Preto is also an EMS, serving an urban area of 127 km$^2$ with about 600,000 inhabitants, located 315 km from the city of São Paulo. The system responds to approximately 4,100 high, medium, and low complexity calls monthly. The system is divided into five regions (or atoms). The ambulances are distributed between these atoms as shown in Fig. 2B.

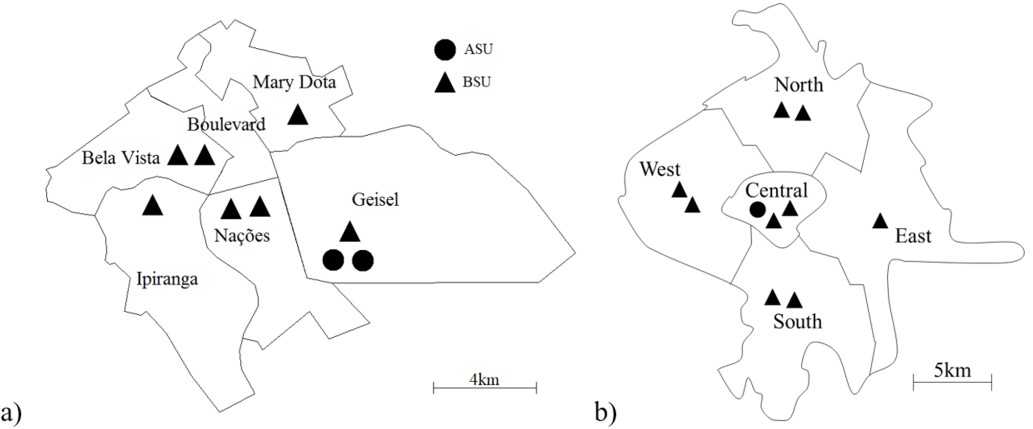

**Figure 2** Maps of the systems studied, showing their ambulance distribution: SAMU-Bauru (A) and SAMU-Ribeirão Preto (B).

In both systems, the classification of calls and dispatch of servers are performed by a physician located in a call center, according to the information provided by the requester at the time of the call. The classification is used to decide on the ambulance dispatch policy, since ASUs (Advanced Service Units) are sent only to calls classified as advanced (high complexity), those which are life-threatening to the patient. SAMU-Bauru has two ASU servers, while SAMU-Ribeirão Preto has only one. Less serious calls are answered by BSUs (Basic Service Units).

In an ECS, calls attended by BSUs are called urgencies and can be divided into more severity classes. For example, in Ribeirão Preto these calls are divided into two classes and in Bauru the urgencies are divided into three classes. For modeling purposes, in this research only two classes of calls were considered: emergencies and urgencies (obtained by the sum of all the divisions considered in the system). The ASU is operated by a team made up of a doctor, a nurse and a driver, and the equipment is more sophisticated than that of the BSU, to provide better care and stabilize patients until they arrive at the hospital. The ASU is sent to a call only when there is an emergency, that is, when illnesses or injuries are life-threatening.

The dispatch policy for servers depends on the spatial distribution and location of the servers. Generally, the servers are decentralized, and the choice of the preferred server is made by analyzing the origin and severity of the call. Preference is given to the server located in the same area (atom) as the call (chosen at random). If all servers in the area are occupied, the first available server closest to the call is chosen. If the call is an emergency, an ASU is immediately dispatched to the site of the incident. If the ASU is busy, the BSU closest to the location of the call is sent. If the call is urgent, the closest available BSU to the location is sent. If all servers are busy, the call is placed in a queue. The call with the highest priority in the queue is serviced by the first available BSU closest to the event. Both ASUs and BSUs can move to any atom.

**Table 3** Demand for the emergency care systems studied.

| | SAMU-Bauru | | | SAMU-Ribeirão Preto | |
|---|---|---|---|---|---|
| **Atom** | **Subatom** | **Arrival rate** | **Atom** | **Subatom** | **Arrival rate** |
| Geisel | 1a | 0.0871 | Central | 1a | 0.0669 |
| | 1b | 0.4876 | | 1b | 0.9140 |
| Nações | 2a | 0.0522 | North | 2a | 0.0469 |
| | 2b | 0.7140 | | 2b | 1.1014 |
| Ipiranga | 3a | 0.0697 | South | 3a | 0.0937 |
| | 3b | 0.3483 | | 3b | 0.7968 |
| Mary Dota | 4a | 0.0348 | East | 4a | 0.0234 |
| | 4b | 0.4179 | | 4b | 0.3281 |
| Bela Vista | 5a | 0.1045 | West | 5a | 0.0469 |
| | 5b | 0.8359 | | 5b | 0.7265 |
| Boulevard | 6a | 0.0174 | | | |
| | 6b | 0.1741 | | | |
| Total | | 3.3435 | Total | | 4.1446 |

It is important to bear in mind that an incoming high priority call will not affect the lower priority calls in progress, characterizing non-preemptive service in both scenarios. Arrival rates were calculated from the proportion of the number of calls from each sub atom to the total number of calls found in the sample. In this way, the total call arrival rate of calls in the system is distributed proportionally to each sub atom.

From the peak demand periods, covering 10 days selected for the two SAMUs, the records of calls were observed in detail, with the objective of estimating a number of variables from the sample for the application and validation of the hypercube model, such as the rate of calls per hour $\lambda$ for each sub atom and the average response time $\mu-1$ of each ambulance (sum of the average travel time with the average service time) (*Souza et al., 2015*; *Takeda, Widmer & Morabito, 2007*).

The values of call arrival rate ($\lambda$) in each sub atom are shown in Table 3; the indices "a" and "b" indicate, respectively, advanced, and basic calls.

The values of the average response time ($\mu-1$) of each server unit (basic and advanced) are shown in Table 4.

As can be seen in Table 4, SAMU-Bauru presents a smaller variation between response times of its ambulances than SAMU-Ribeirão Preto, although the average service time in Bauru is longer. It is also noted that the service provided by ASU units requires more time than those of BSU.

In addition, the average journey time of an ambulance to each atom in the city or within the same atom is a measure to be observed in each configuration, considered an input for the Hypercube model. Table 5 presents these data for SAMU-Bauru and SAMU-Ribeirão Preto, in which the journey time pattern is similar for both SAMUs. As journeys between any points within and between regions were considered, it is possible that the average journey time within an atom is greater than the journey time from that atom to another, considering the distances covered, and geographical and traffic aspects. An ambulance

**Table 4 Average response times for each server unit.**

| Server type | Ambulances | Average response time (min.) | |
|---|---|---|---|
| | | **Bauru** | **Ribeirão Preto** |
| ASU | 1 | 55.5 | 47.2 |
| | 2 | 55.5 | – |
| BSU | 1 | 46.0 | 30.4 |
| | 2 | 38.1 | 30.4 |
| | 3 | 40.9 | 45.6 |
| | 4 | 45.5 | 45.6 |
| | 5 | 46.7 | 33.1 |
| | 6 | 46.7 | 33.1 |
| | 7 | 43.3 | 38.3 |
| | 8 | – | 39.6 |
| | 9 | – | 39.6 |
| Average | ASU | 55.5 | 47.2 |
| | BSU | 43.9 | 37.3 |
| | Total | 46.5 | 38.3 |

**Table 5 Average journey times between atoms for the SAMUs in Bauru and Ribeirão Preto.**

| | Bauru | | | | | | Ribeirão Preto | | | | |
|---|---|---|---|---|---|---|---|---|---|---|---|
| **Atom** | **1** | **2** | **3** | **4** | **5** | **6** | **Atom** | **1** | **2** | **3** | **4** | **5** |
| **1** | 8.2 | 13.7 | 13.2 | 9.9 | 10.0 | 7.6 | **1** | 8.7 | 9.5 | 10.9 | 11.4 | 9.9 |
| **2** | 13.7 | 11.7 | 15.0 | 16.0 | 15.0 | 9.0 | **2** | 9.5 | 11.6 | 9.7 | 7.3 | 7.5 |
| **3** | 13.2 | 15.0 | 8.7 | 15.3 | 17.0 | 11.0 | **3** | 10.9 | 9.7 | 8.6 | 13.5 | 10.7 |
| **4** | 9.9 | 16.0 | 15.3 | 10.8 | 15.0 | 7.4 | **4** | 11.4 | 7.3 | 13.5 | 10.1 | 15.1 |
| **5** | 10.0 | 15.0 | 17.0 | 15.0 | 8.8 | 11.0 | **5** | 9.9 | 7.5 | 10.7 | 15.1 | 9.2 |
| **6** | 7.6 | 9.0 | 11.0 | 7.4 | 11.0 | 7,6 | – | – | – | – | – | – |

could take longer to cross an atom than to answer a call at a point in another atom near the location of the ambulance.

In addition to all of the above, it is worth mentioning that statistical tests were also conducted in order to verify whether the characteristics of the SAMUs in Bauru and Ribeirão Preto fit the nine assumptions necessary for the application of the hypercube model. After these tests were performed, the application of the hypercube model was validated for both cases. Details of these tests can be seen in *Souza et al. (2015)* and *Beojone & Souza (2017)*.

With the cases duly presented, in the next four sections the CI construction method steps are presented.

## Determination of the configurations to be compared (DMU determination)

The elaboration of definitive configurations to be compared was based on the permutation of all possible combinations of ambulance distributions in different atoms. From the

concepts of partition and combinatorial analysis, it was found that, in the case of SAMU-Bauru, considering that there are two ASU servers and seven BSU servers across six different atoms, a number of 16,632 initial configuration possibilities for the system can be built. For SAMU-Ribeirão Preto, the number of combinations is lower, since there is one ASU server and nine BSU servers across just five atoms, which generates a number of 3,575 combinations.

In furtherance of making the application of the hypercube model feasible, in order to obtain the desired inputs and outputs, the performance measures of the hypercube model for all possible locations of the BSUs were calculated, giving 792 configurations for Bauru and 715 for Ribeirão Preto, assuming that the ASUs remained within their original atoms. This simplification, where the ASUs are kept to one atom, can also be justified by the fact that, according to the current system organization, only the ambulance service centers can manage the receipt of advanced calls. It is worth mentioning that the same servers, when located at the same base, are considered homogeneous.

## Application of the hypercube model to the determination of inputs and outputs

Before applying the hypercube model, it was necessary to calculate model input data from the sample data, in this case, the service fees of the servers and the dispatch preference matrix. The calculation of service rates depends on two factors: the service time and the journey time. The service time was obtained from the data sample collected at SAMU. A mean value was considered for all ASUs from the sample. On the other hand, journey times depend on the location of the servers. For this, as in the original scenario there are servers in all atoms, the hypercube model was applied, and the initial journey times used in calculating the service fees of the servers were obtained.

The establishment of a dispatch preference policy depends on several factors, such as: the type of call, the location of the call, the location of the servers, and the presence of randomness in the dispatch. Therefore, ASUs prefer answering emergency calls but are not sent for urgent calls. For emergency calls, The BSUs are sent as backups to the ASUs, and the first backup is the BSU located in the atom of the occurrence, the others being chosen at random. For urgent calls, the BSU located in the atom of the occurrence has the dispatch preference, the others are chosen randomly, as a backup.

For the analysis of the configurations, the hypercube model with no queue (*Iannoni, Morabito & Saydam, 2008*) and with a waiting queue with priority service discipline (*Beojone, Máximo de Souza & Iannoni, 2020*), were considered. In addition, the possibility of a demand increase of 25% and 50% over the original arrival rate of all atoms was considered. When adopting the classification of calls (emergency and urgency) and randomness in the choice of servers, the model of *Takeda, Widmer & Morabito (2007)*, the layering technique and generation of random dispatch preference matrices was used. Response times were calibrated for the different configurations.

Finally, after solving the hypercube model for each configuration, the workloads and response times of the servers were recorded, as well as the response times for atoms, which

**Table 6   The most efficient configurations in Bauru and Ribeirão Preto.**

| General ranking | Configuration ID | Bauru | | | | | | | | | | | |
|---|---|---|---|---|---|---|---|---|---|---|---|---|---|
| | | Number of ambulances | | | | | | Ranking in each scenario | | | | | |
| | | A1 | A2 | A3 | A4 | A5 | A6 | P0pq | P0nq | P25pq | P25nq | P50pq | P50nq |
| 1 | "6" | 7 | 0 | 0 | 0 | 0 | 0 | 6 | 1 | 1 | 1 | 1 | 1 |
| 2 | "9" | 0 | 0 | 0 | 1 | 0 | 6 | 2 | 2 | 2 | 3 | 2 | 47 |
| 3 | "183" | 0 | 1 | 0 | 2 | 0 | 4 | 8 | 3 | 3 | 2 | 3 | 16 |
| 4 | "13" | 0 | 0 | 1 | 0 | 0 | 6 | 1 | 4 | 4 | 9 | 4 | 96 |
| 5 | "165" | 0 | 0 | 1 | 2 | 0 | 4 | 12 | 5 | 5 | 5 | 5 | 44 |
| 6 | "72" | 0 | 0 | 1 | 1 | 0 | 5 | 14 | 6 | 6 | 6 | 6 | 33 |
| | | Ribeirão Preto | | | | | | | | | | | |
| 1 | "36" | 0 | 7 | 0 | 2 | 0 | – | 1 | 1 | 1 | 1 | 1 | 1 |
| 2 | "86" | 0 | 6 | 0 | 3 | 0 | – | 2 | 2 | 2 | 2 | 2 | 2 |
| 3 | "16" | 0 | 8 | 0 | 1 | 0 | – | 3 | 3 | 3 | 3 | 3 | 3 |
| 4 | "119" | 0 | 6 | 2 | 1 | 0 | – | 4 | 4 | 4 | 4 | 4 | 4 |
| 5 | "83" | 0 | 3 | 0 | 6 | 0 | – | 5 | 5 | 5 | 5 | 6 | 6 |
| 6 | "392" | 0 | 2 | 4 | 3 | 0 | – | 6 | 6 | 6 | 6 | 7 | 7 |

are necessary for the calculation of the inputs and outputs. In addition, the probabilities of loss and the system being empty were also recorded, as a way of verifying the model

## DEA/BoD application (normalization, weighting, and aggregation)

To rank the configurations, it is assumed that the configuration with the best performance should allow a low response time for each atom and each server (outputs), without unduly increasing the workload of each server (inputs).

Thus, as inputs, the workload of each BSU server was used; that is, seven inputs in the application for Bauru, and nine in the application for Ribeirão Preto. It is worth mentioning that as the ASU servers are dedicated to their atoms, their workload did not vary from configuration to configuration.

The outputs used were the reciprocal of the response time for each sub atom, and the reciprocal of the response time of each BSU server, so that the response time of the ASU servers was again disregarded. Thus, the model for SAMU-Bauru considered 19 outputs (12 relating to the sub atoms and seven relating to the BSUs) and the model for SAMU-Ribeirão Preto considered 19 outputs (10 relating to the sub atoms and nine relating to the BSUs). It is noteworthy that, even with the excess of variables, the number of DMUs generated was more than enough to guarantee good discrimination (*Cooper, Seiford & Tone, 2007*).

To complement the analysis, different rankings were obtained for the six scenarios, with distinct levels of demand and queuing policies. Tables 6 and 7 show the best and worst configurations, respectively, in Bauru and Ribeirão Preto for each scenario. The acronyms in the table represent: (a) P0pq - current demand with permitted queue; (b) P0nq - current demand with no queue allowed; (c) P25pq - 25% higher demand with queue; (d) P25nq - 25% higher demand with no queue; (e) P50pq - 50% higher demand with queue; (f) P50nq - 50% higher demand with no queue.

**Table 7  The least efficient configurations in the cities of Bauru and Ribeirão Preto.**

| General ranking | Configuration ID | Bauru | | | | | | | | | | | |
|---|---|---|---|---|---|---|---|---|---|---|---|---|---|
| | | Number of ambulances | | | | | | Ranking in each scenario | | | | | |
| | | A1 | A2 | A3 | A4 | A5 | A6 | P0pq | P0nq | P25pq | P25nq | P50pq | P50nq |
| 792 | "3" | 0 | 0 | 0 | 7 | 0 | 0 | 792 | 792 | 792 | 792 | 792 | 792 |
| 791 | "1" | 0 | 0 | 0 | 0 | 0 | 7 | 791 | 791 | 791 | 791 | 791 | 791 |
| 790 | "4" | 0 | 0 | 7 | 0 | 0 | 0 | 790 | 789 | 789 | 788 | 789 | 786 |
| 789 | "60" | 2 | 0 | 5 | 0 | 0 | 0 | 789 | 787 | 788 | 786 | 788 | 787 |
| 788 | "150" | 3 | 0 | 4 | 0 | 0 | 0 | 788 | 790 | 790 | 790 | 790 | 784 |
| 787 | "149" | 3 | 0 | 0 | 4 | 0 | 0 | 787 | 786 | 787 | 787 | 787 | 789 |
| | | Ribeirão Preto | | | | | | | | | | | |
| 715 | "3" | 0 | 0 | 9 | 0 | 0 | – | 715 | 715 | 715 | 715 | 715 | 715 |
| 714 | "20" | 1 | 0 | 8 | 0 | 0 | – | 714 | 714 | 714 | 714 | 714 | 714 |
| 713 | "10" | 0 | 0 | 8 | 0 | 1 | – | 713 | 713 | 713 | 713 | 713 | 713 |
| 712 | "5" | 9 | 0 | 0 | 0 | 0 | – | 712 | 712 | 712 | 712 | 712 | 712 |
| 711 | "40" | 2 | 0 | 7 | 0 | 0 | – | 711 | 711 | 711 | 711 | 711 | 711 |
| 710 | "18" | 1 | 0 | 0 | 0 | 8 | – | 710 | 710 | 710 | 710 | 710 | 710 |

The best configuration for SAMU-Bauru was, almost invariably, to concentrate all seven ambulances on atom A1 (Nações), which is the atom that occupies a central position, as can be seen in Fig. 2A. The exception was the current demand situation with queue (P0pq), in which the best configuration was to place six ambulances on atom A6 (Boulevard) and one on atom A3 (Ipiranga). For Ribeirão Preto, in all situations, the optimal configuration was to place seven BSUs in atom A2 (South) and two in atom A4 (West). For both cities, and in all situations analyzed, the best configurations involved concentrating ambulances on a few atoms.

## Composite index and ranking analysis (validation)

To establish patterns and explain what makes the configurations more efficient, Spearman's correlation coefficient was determined, as it is indicated for cases in which ordinal data is used. By this means, the relationship between the number of ambulances in each atom and the efficiency ranking in each of the situations proposed was assessed, in addition to investigating the differences between the rankings obtained. Considering the correlation between the rankings for the two cities, all correlations were significant, being above 94%.

Table 8 shows Spearman's correlation coefficients in terms of the number of ambulances in each atom and the ranking obtained in each situation proposed.

As can be seen, for all the situations proposed in Atom A1 (Nações), there is a positive correlation between the ranking of configurations and the number of ambulances. Therefore, the more ambulances allocated to this atom, the less efficient the configuration tends to be. On the other hand, with respect to atoms A6 (Boulevard) and A2 (Geisel), the result was exactly the opposite: the higher the concentration of ambulances, the better positioned the configuration in the efficiency ranking. This was seen most acutely at atom A6. For the other atoms, the correlations obtained were low or insignificant. The correlation

**Table 8** Correlation matrix considering the number of ambulances in each atom and the rankings for each configuration in Bauru.

| Atom | P0pq | P0nq | P25pq | P25nq | P50pq | P50nq |
|---|---|---|---|---|---|---|
| A1 | 0.42*** | 0.46*** | 0.45*** | 0.43*** | 0.44*** | 0.44*** |
| A2 | −0.14*** | −0.15*** | −0.12*** | −0.11*** | −0.09*** | −0.09*** |
| A3 | −0.01 | −0.08** | −0.05 | −0.03 | −0.01 | 0.05 |
| A4 | −0.02 | −0.12*** | −0.09*** | −0.07* | −0.06 | 0.05 |
| A5 | 0.01 | 0.08** | 0.07** | 0.05 | 0.04 | 0.00 |
| A6 | −0.61*** | −0.54*** | −0.61*** | −0.62*** | −0.67*** | −0.79*** |
| Standard deviation | 0.35*** | 0.35*** | 0.37*** | 0.35*** | 0.36*** | 0.37*** |

Notes.
One asterisk (*) represents a significance level of 0.05, indicating a 95% confidence in the observed relationship. Two asterisks (**) mean a significance level of 0.01, or 99% confidence. Three asterisks (***) indicate the highest level of significance, at 0.001, or 99.9% confidence.

**Table 9** Correlation matrix considering the number of ambulances in each atom and the rankings for each configuration in Ribeirão Preto.

| Atom | P0pq | P0nq | P25pq | P25nq | P50pq | P50nq |
|---|---|---|---|---|---|---|
| A1 | 0.14*** | 0.14*** | 0.14*** | 0.15*** | 0.15*** | 0.13*** |
| A2 | −0.18*** | −0.17*** | −0.18*** | −0.18*** | −0.18*** | −0.17*** |
| A3 | 0.10*** | 0.10*** | 0.10*** | 0.10*** | 0.12*** | 0.11*** |
| A4 | −0.20*** | −0.21*** | −0.19*** | −0.20*** | −0.20*** | −0.20*** |
| A5 | 0.13*** | 0.14*** | 0.13*** | 0.14*** | 0.12*** | 0.13*** |
| Standard deviation | 0.34*** | 0.35*** | 0.35*** | 0.35*** | 0.35*** | 0.35*** |

Notes.
One asterisk (*) represents a significance level of 0.05, indicating a 95% confidence in the observed relationship. Two asterisks (**) mean a significance level of 0.01, or 99% confidence. Three asterisks (***) indicate the highest level of significance, at 0.001, or 99.9% confidence.

between rankings and the standard deviation of the number of ambulances in atoms was also analyzed, to evaluate the hypothesis that the configurations in which ambulances are more concentrated on certain atoms tend to be more efficient. As shown in Table 8, contrary to what was expected, the concentration of ambulances contributes negatively to the efficiency of the configurations.

Although atoms A1 and A6 occupy central positions geographically, the results with respect to the concentration of ambulances were opposing. It is concluded, then, that there must be other variables that explain the rankings, in addition to atom centrality.

For SAMU-Riberão Preto, all correlations between the number of servers in each atom and the efficiency ranking were significant, as shown in Table 9.

The configurations for the ambulances concentrated on atoms A2 (South) and A4 (West) were the most efficient. In all others, including the central (A5) atom, the presence of ambulances tends to worsen the level of efficiency. It is noteworthy that atoms A2 and A4 are adjacent, situated in the western part of Ribeirão Preto. It should also be noted that the correlation coefficients were lower than those obtained for the SAMU-Bauru, varying from 0.1 to 0.2.

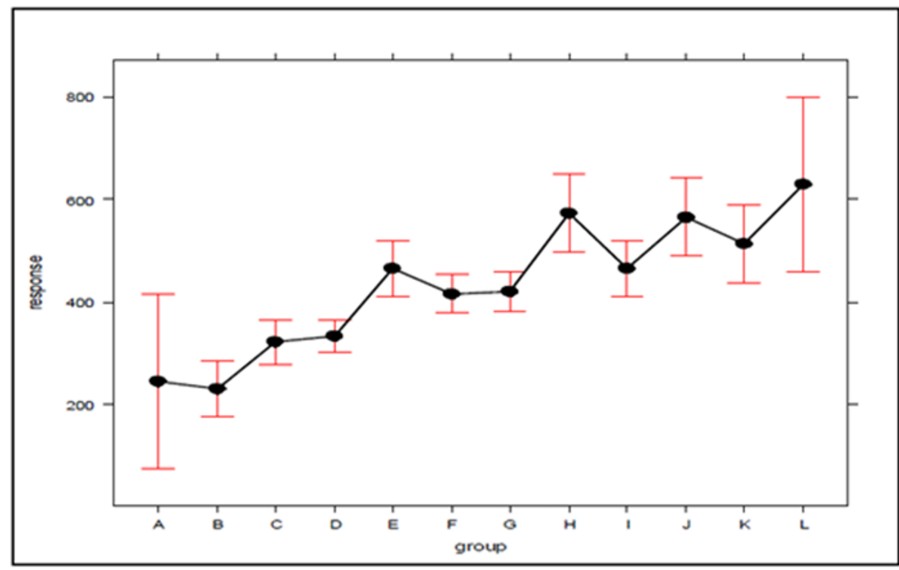

**Figure 3** **The effect of ambulance concentration on the SAMU-Bauru efficiency ranking.**

Comparing the two cities, the most evident similarity was in relation to the correlation between the standard deviation of the number of ambulances and the efficiency ranking. For Ribeirão Preto, a lower concentration was also related to greater efficiency. It should be noted, however, that the objective here is not to generalize these patterns, but to demonstrate the usefulness of the method to explain them in each emergency system.

In the search for new explanations, main effects plots were created, given in Figs. 3 and 4, which relate the position of an atom in the efficiency ranking (response variable) with respect to the concentration of ambulances (which can be expressed by the standard deviation) for Bauru and Ribeirão Preto, respectively. In these plots, the first group (A for both cities) encompasses the configurations in which the servers are fully distributed between atoms, and the second group (L for Bauru and Q for Ribeirão Preto) encompasses the configurations in which the ambulances are concentrated in a single atom, with the other groups representing intermediate concentration levels. It should be noted that these plots were generated using an attachment developed for Excel. To generate the graphs, the results regarding the current demand with queue, which represents the current situation, were considered.

As can be seen in Figs. 3 and 4, there is a clear tendency for the ranking of configurations to worsen, the higher the level of concentration. Worthy of note is the fact that the groups with the lowest (A) and highest concentrations (L in Figs. 3 and Q in 4) are those with the greatest dispersion of results, encompassing both high and low efficiency configurations. As average response time indicators were adopted for the system, the results described here are not able to express differences according to the geographic distribution of occurrences. For example, it is not possible to say to what extent the concentration of ambulances at a given atom has worsened performance in areas far from that atom, or improved service in

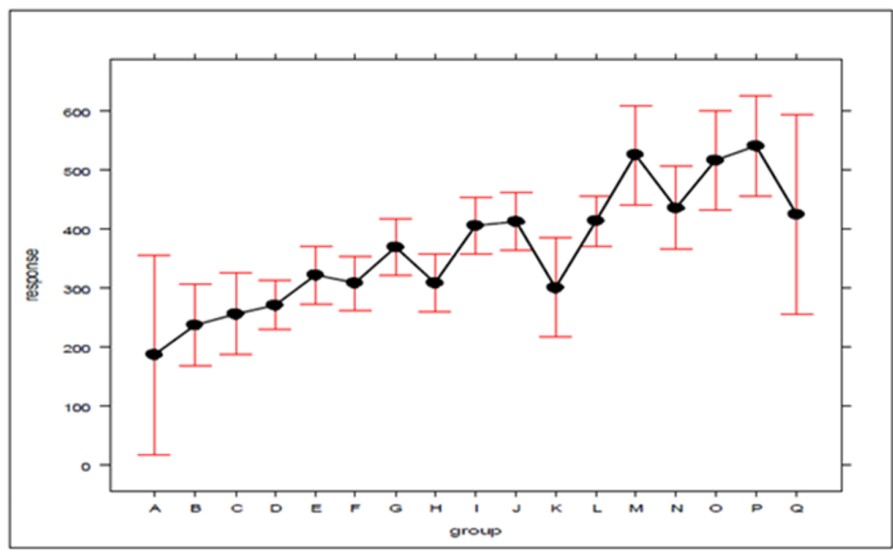

**Figure 4   The effect of ambulance concentration on the SAMU-Ribeirão Preto efficiency ranking.**

areas close to it. The standards found must be validated by decision-makers, in terms of their technical and economic feasibility, to be effectively implemented.

## CONCLUSIONS

The objective of this article was to propose a new composite index (CI) to identify an effective configuration for emergency care in situations where there are multiple performance indicators that present trade-offs.

The conclusion drawn from the proposed integration was successful, and that, as by-products of this work, the following can be highlighted: (a) a standardized procedure for generating comparison configurations based on combinatorial analysis and the concept of partition; (b) a computational model for implementing the hypercube model, determining relatively quickly all parameters for the configurations generated; (c) the determination of inputs and outputs to be used to evaluate the performance of SAMUs, as well as the establishment of an adequate DEA model, based on BCC models, inverted frontier and weight constraints, which guaranteed good discrimination for the configurations; and (d) effectively testing the proposed model using real data collected from SAMU operators in Bauru and Ribeirão Preto.

Emergency care system managers may benefit from the tool presented, both in the system design phase and during its operation, conducting periodic assessments of the efficiency of the configurations as new data is collected. The solution to the challenge presented in the article - increasing the efficiency of emergency systems - has a significant social impact: providing better quality services at a reduced cost to society.

The analysis procedure presented here emphasizes the particularities of the emergency systems of the cities studied, which must be taken into consideration. While the proposed DEA-hypercube integration model does have general application, incorporating the

variables and indicators usually adopted when problems of locating emergency systems are considered, it is suggested that the model be adapted to distinct levels of priority calls, dispatch policies and other regional service divisions.

As a recommendation for future studies, pattern analysis could be further developed through the adoption of other statistical tools, such as factorial analysis, so that the patterns identified in the systems studied are even clearer. In addition, the integrated model presented here could be applied to other emergency systems (police, firefighters), as well as in cities with different demographic profiles and emergency systems with characteristics at variance to the cases presented. Furthermore, modifications to the DEA model proposed could be considered, such as the use of fuzzy DEA to incorporate potential uncertainties in the inputs and outputs. Other techniques to increase discrimination between efficient configurations could also be applied, such as cross-evaluation and the triple index. Finally, issues related to the usability of the model by managers and employees could be further investigated.

### Funding
This work was supported by The São Paulo Research Foundation, FAPESP (No. 14/19051-7). The funders had no role in study design, data collection and analysis, decision to publish, or preparation of the manuscript.

### Grant Disclosures
The following grant information was disclosed by the authors:
The São Paulo Research Foundation, FAPESP: No. 14/19051-7.

### Competing Interests
The authors declare there are no competing interests.

### Author Contributions
- Enzo Barberio Mariano conceived and designed the experiments, performed the experiments, analyzed the data, performed the computation work, prepared figures and/or tables, and approved the final draft.
- Regiane Máximo Siqueira conceived and designed the experiments, performed the experiments, analyzed the data, performed the computation work, prepared figures and/or tables, and approved the final draft.
- Caio Vitor Beojone performed the experiments, analyzed the data, performed the computation work, prepared figures and/or tables, and approved the final draft.
- Octaviano Rojas Luiz analyzed the data, performed the computation work, prepared figures and/or tables, authored or reviewed drafts of the article, and approved the final draft.
- João Victor Rojas Luiz analyzed the data, performed the computation work, prepared figures and/or tables, authored or reviewed drafts of the article, and approved the final draft.

**Peer**J Computer Science

- Hugo Henrique dos Santos analyzed the data, performed the computation work, prepared figures and/or tables, authored or reviewed drafts of the article, and approved the final draft.

## Data Availability

The data and code are available at Zenodo: Enzo Barberio Mariano, Regiane Máximo Siqueira, Caio Vitor Beojone, Octaviano Rojas Luiz, João Victor Rojas Luiz, & Hugo Henrique dos Santos. (2023). Computer Code and Data - Determination of server location in emergency care systems: an index proposal using Data Envelopment Analysis and the Hypercube Queuing Model [Data set]. Zenodo. https://doi.org/10.5281/zenodo.8179254.

## Supplemental Information

Supplemental information for this article can be found online at http://dx.doi.org/10.7717/peerj-cs.1637#supplemental-information.

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
