# Peer review of "Determination of server location in emergency care systems: an index proposal using data envelopment analysis and the hypercube queuing model"

_PeerJ Computer Science, doi:10.7717/peerj-cs.1637_

## Round 0.1 · original submission · Major Revisions

Dear authors,

Your paper has been reviewed by two experts who recommended major revisions. Please correct your paper and provide cover letter with replies point to point to each comment.

**Language Note:** The review process has identified that the English language must be improved. PeerJ can provide language editing services - please contact us at copyediting@peerj.com for pricing (be sure to provide your manuscript number and title). Alternatively, you should make your own arrangements to improve the language quality and provide details in your response letter. – PeerJ Staff

Reviewer 1 ·

Basic reporting

To make the paper even better, I give certain suggestions:
The paper is written on a large number of pages. The editor makes a decision on the scope of the paper.
In some parts of the paper, the first person plural is used to write the paper (line 172, line 209, line 311 & line 677). That needs to be corrected. The paper can not be written in the first person!
The introduction of the paper is extensive (≈ 3 pages). I suggest that the authors divide the introduction into 2 chapters (1. Introduction and 2. literature review), or to shorten the introduction.
At the end of the introduction, it is necessary to explain the rest of the paper by chapter (2-3 sentences)
Shorten the conclusion. In the conclusion, there are also redundant sentences.

Experimental design

It is a very interesting research, based on hypercube queuing model... The paper represents an extremely important segment of operational research. The authors approached the paper with extremely precise methodological approach (the main contributions of this paper – to propose a new composite index (CI) that helps to determine the most effective location of servers in an Emergency Care System (ECS)). The abstract of the paper describes the problem extremely well and indicates the objectives. The authors present an excellent literature review.
Special praise to the author for the methodology of the paper and the given case study.

Validity of the findings

No comment

Additional comments

No comment

·

Basic reporting

The article titled "Determination of server location in emergency care systems: an index proposal using Data Envelopment Analysis and the Hypercube Queuing Model" presents a valuable contribution to the field of emergency care systems. The proposed composite index (CI) offers a novel approach to determining the optimal server location. While the article demonstrates a good contribution, there are areas that require improvement. This review report aims to provide constructive feedback to enhance the clarity and quality of the article.
[1] Emphasizing the Contribution:
The authors should focus on highlighting the significance and unique aspects of their contribution in a more explicit manner. It would be beneficial to clearly state the precise problem or research gap the article addresses and how the proposed CI using DEA/BoD and the Hypercube Queuing Model adds value to existing literature on server location determination in emergency care systems.
[2] Introduction Section:
The introduction section needs improvement to provide a comprehensive overview of the research area and establish the context for the study. The authors should consider expanding on the background information, such as the importance of server location in emergency care systems and the challenges associated with it. Additionally, providing a brief summary of the existing methods and their limitations would help readers understand the motivation behind the proposed CI.
[3] Literature Review:
The literature review should be updated to incorporate relevant and recent studies in the field. I suggest some articles below:
Taletović, M., & Sremac, S. (2023). PCA-DEA model for efficiency assessment of transportation company. International Journal of Management and Decision Making, 2(1), 11-20.
Blagojević, A., Vesković, S., Kasalica, S., Gojić, A., & Allamani, A. (2020). The application of the fuzzy AHP and DEA for measuring the efficiency of freight transport railway undertakings. Operational Research in Engineering Sciences: Theory and Applications, 3(2), 1-23.

Experimental design

no comment

Validity of the findings

no comment

---

## Round 0.2 · accepted · Accept

Dear authors,

Your revised version has been accepted by both reviewers.

Reviewer 1 ·

Basic reporting

No comment

Experimental design

No comment

Validity of the findings

No comment

Additional comments

No comment

·

Basic reporting

The necessary corrections have been made, and the paper is now in a state where it can be accepted for publication.

Experimental design

The necessary corrections have been made, and the paper is now in a state where it can be accepted for publication.

Validity of the findings

The necessary corrections have been made, and the paper is now in a state where it can be accepted for publication.

Additional comments

The necessary corrections have been made, and the paper is now in a state where it can be accepted for publication.